# Disease mapping: Geographic differences in population rates of interventional treatment for prostate cancer in Australia

Jessica K. Cameron[1,2]*, Upeksha Chandrasiri[2], Jeremy Millar[3], Joanne F. Aitken[2,4,5], Susanna Cramb[1,6], Jeff Dunn[7,8], Mark Frydenberg[9], Prem Rashid[10], Kerrie Mengersen[1], Suzanne K. Chambers[11,12], Peter D. Baade[1,2,12], David P. Smith[13,14]

1 Centre for Data Science, Queensland University of Technology, Brisbane, Queensland, Australia, 2 Cancer Council Queensland, Spring Hill, Queensland, Australia, 3 Central Clinical School, Monash University, Clayton, Victoria, Australia, 4 School of Public Health, The University of Queensland, Herston, Queensland, Australia, 5 School of Public Health and Social Work, Queensland University of Technology, Kelvin Grove, Queensland, Australia, 6 Australian Centre for Health Services Innovation (AusHSI) and Centre for Healthcare Transformation, Queensland University of Technology, Kelvin Grove, Queensland, Australia, 7 Prostate Cancer Foundation Australia, St Leonards, New South Wales, Australia, 8 Institute for Resilient Regions, University of Southern Queensland, Springfield Central, Queensland, Australia, 9 Department of Surgery, Monash University, Clayton, Victoria, Australia, 10 Medicine & Health, University of New South Wales, Randwick, New South Wales, Australia, 11 Health Sciences, Australian Catholic University, Banyo, Queensland, Australia, 12 Menzies Health Institute Queensland, Griffith University, Southport, Queensland, Australia, 13 The Daffodil Centre, The University of Sydney, a Joint Venture with Cancer Council NSW, Sydney, New South Wales (NSW), Australia, 14 School of Public Health and Preventive Medicine, Monash University, Melbourne, Victoria, Australia

* jessicacameron@cancerqld.org.au

**Data Availability Statement:** All relevant data are within the paper and its Supporting information files.

## Abstract

### Background

Treatment decisions for men diagnosed with prostate cancer depend on a range of clinical and patient characteristics such as disease stage, age, general health, risk of side effects and access. Associations between treatment patterns and area-level factors such as remoteness and socioeconomic disadvantage have been observed in many countries.

### Objective

To model spatial differences in interventional treatment rates for prostate cancer at high spatial resolution to inform policy and decision-making.

### Methods

Hospital separations data for interventional treatments for prostate cancer (radical prostatectomy, low dose rate and high dose rate brachytherapy) for men aged 40 years and over were modelled using spatial models, generalised linear mixed models, maximised excess events tests and k-means statistical clustering.

**Funding:** This work was supported by an Australian Research Council Linkage grant (LP200100468) granted to KM, PB, JA and SC. https://www.arc.gov.au/ This work was also supported by a Centre for Research Excellence grant from The National Health and Medical Research Council, Australia (1116334) granted to SKC, JD, DS, PB, JA and SC. https://www.nhmrc.gov.au/ The funders had no role in study design, data collection and analysis, decision to publish, or preparation of the manuscript.

**Competing interests:** The authors have declared that no competing interests exist.

## Results

Geographic differences in population rates of interventional treatments were found (p<0.001). Separation rates for radical prostatectomy were lower in remote areas (12.2 per 10 000 person-years compared with 15.0–15.9 in regional and major city areas). Rates for all treatments decreased with increasing socioeconomic disadvantage (radical prostatectomy 19.1 /10 000 person-years in the most advantaged areas compared with 12.9 in the most disadvantaged areas). Three groups of similar areas were identified: those with higher rates of radical prostatectomy, those with higher rates of low dose brachytherapy, and those with low interventional treatment rates but higher rates of excess deaths. The most disadvantaged areas and remote areas tended to be in the latter group.

## Conclusions

The geographic differences in treatment rates may partly reflect differences in patients' physical and financial access to treatments. Treatment rates also depend on diagnosis rates and thus reflect variation in investigation rates for prostate cancer and presentation of disease. Spatial variation in interventional treatments may aid identification of areas of under-treatment or over-treatment.

## Introduction

Prostate cancer is the most common cancer diagnosed among Australian men [1], with an estimated 24 217 new cases in 2022 [1]. Age-standardised incidence rates have increased to 151 cases per 100,000 persons-years in 2022 from 80 cases per 100,000 person-years in 1982 [1]. Survival has improved from 61% (5-year relative survival) in the late 1980s to over 95% in 2013–2017 [1]. Incidence and survival vary geographically, and in Australia, men living in remote areas have lower incidence but poorer survival compared to men living in urban areas [2]. This could be related to differences in testing rates [3] or differential access to and use of health services in the management of prostate cancer [4].

There is no single "best" treatment for prostate cancer [5]. Depending on the circumstances, management options may include interventional treatments requiring hospital admission and surgery (radical prostatectomy, low dose rate (LDR) brachytherapy or high dose rate (HDR) brachytherapy), or less invasive treatments such as active surveillance [6]. Treatment decisions are influenced by many factors, including the stage and grade of cancer [7], personal preference [7], general health [7], age at diagnosis [8], potential side effects or complications [9], proximity of residence to treatment facilities [7] and whether patients have private health insurance [8]. While radical prostatectomy remains the most frequent treatment for localised prostate cancer, there is increasingly a preference for less invasive treatments for men with low-risk disease [6].

Treatment rates for prostate cancer at any stage have been shown to vary by residential remoteness and area-level socioeconomic disadvantage [4]. For example, in 2008–2013, men residing in regional areas of the state of Victoria were 47% less likely to receive any curative treatment compared to those in urban areas [10] although these patterns may have changed over time [11]. Additionally, lower rates of radical prostatectomy were observed among men living in more socioeconomically disadvantaged areas than in more advantaged areas [4].

Disease mapping is increasingly used to help understand the use of and need for services at the small area level [12]. There is a lack of information about how interventional treatment patterns for prostate cancer vary at a finer granularity, by small geographical area, in Australia. In

this study we addressed this gap using population-level data sets to compare hospital separation rates for interventional treatments for prostate cancer. We tested the hypothesis that rates varied by small area and provided novel insights into how these vary depending on where patients live.

## Materials and methods

We used population data from hospital administrative databases and various statistical models to examine geographic variation in population-level separation rates for radical prostatectomy, LDR brachytherapy and HDR brachytherapy.

### Data

We obtained counts of all public and private hospital separations in Australia for males aged 40 years and over having radical prostatectomy, LDR brachytherapy or HDR brachytherapy for the treatment of prostate cancer, using the Australian Classification of Health Intervention (ICD-10-AM/ACHI 6[th] edition) refined Diagnosis Related Groups codes (see S5 Table in S1 Appendix). Data were obtained from the Australian Institute of Health and Welfare's National Hospital Morbidity Database in August 2019. As inpatient treatments, population-level data is available for these treatments since all occasions are recorded in hospital separations data. Separations were not obtained for records in which age or sex were not reported, if the care type was "newborn", for posthumous organ procurement and for hospital boarders.

Data were provided aggregated by procedure code, 10-year age group, financial year (defined as July 1 to June 30) and area of residence at time of treatment, for separations between July 2006 and June 2017. Data for financial years 2006–7 to 2011–12 were coded using Statistical Local Areas from the 2006 Australian Standard Geographical Classification [13] and for 2012–13 to 2016–17 using Statistical Area Level 2 (SA2) from the 2011 Australian Statistical Geographical Standard [14]. Separations coded as "migratory", "offshore", "shipping", "no usual address" or "undefined" were excluded from all analyses. The remote islands of Christmas Island, Cocos Island and Lord Howe Island were excluded from the spatial modelling, as were SA2s with an average annual male population aged 40 or over that was less than 5. Administrative data were used for this study. Individuals were not recruited and the requirement for individual consent was waived by the Ethics Committee. The aggregated data were considered to be potentially re-identifiable.

Estimated male resident population data by SA2, calendar year and 5-year age group [15], along with area-based measures of remoteness and the Index of Relative Socioeconomic Advantage and Disadvantage were obtained from Australian Bureau of Statistics [16, 17].

To describe ecological correlations between rates of interventional treatment, incidence and survival among men diagnosed with prostate cancer, publicly available modelled standardised incidence ratios (SIR, the relative risk in diagnoses) and excess hazard ratios (EHR, the relative risk of death in excess of population mortality) for prostate cancer for each area in 2012–2016 (SIRs) or 2007–2016 (EHRs) were obtained from the Australian Cancer Atlas [2].

**Preparation of the data.** Count data geocoded by Statistical Local Area were randomly reallocated to SA2s proportionately using the correspondence matrices obtained from the Australian Bureau of Statistics (see supplementary material for details) [18]. Similarly, counts were reallocated from financial to calendar year proportionately to population size.

### Statistical modelling

**Software.** Data processing, plotting and clustering were conducted using R (v4.0.0) and the CARBayes package (v5.1) was used for spatial modelling. Area-level associations were modelled using Stata (v16.0).

**Area-level associations.** Generalised linear models were built to model area-level count data between 2007 and 2016 for each procedure with the following covariates: age group, State or Territory, remoteness category and area-level socioeconomic quintile. All covariates were retained in the multivariable model. Likelihood ratio tests were used initially to test for univariable associations and to determine the strength of evidence for the association in the final, fully-adjusted model. Poisson and negative binomial models were trialled, as were mixed effects models with a random effect for SA2. Based on the results of likelihood ratio tests as well as the AIC and BIC, mixed effects negative binomial models were fitted for radical prostatectomy and LDR brachytherapy and a mixed effects Poisson model for HDR brachytherapy.

**Spatial modelling.** Bayesian spatial modelling was undertaken using the methods described for the Australian Cancer Atlas, with details provided in the Supplementary Materials [19]. Briefly, counts of separations in the period 2007 to 2016 for each treatment were modelled separately, offset by the age-adjusted expected counts.

A spatial random effect was included in the models to adjust the separation rates for each area based on its neighbours' rates. The model provided a set of spatially smoothed area-level standardised separation rate ratios (SSRs) relative to the national average.

Spatial heterogeneity was tested using Tango's Maximised Excess Events Test [20].

To characterise areas according to their separation rates across all treatments considered in this study, the areas were statistically grouped using *k*-means clustering [21]. SIRs and EHRs were also included in *k*-means clustering since population-based SSRs are associated with prostate cancer incidence (SIRs), and survival (EHRs) is an indicator of area-level differences in disease progression. Statistical clustering grouped areas so that covariates (SSRs, SIRs and EHRs) within each group were as similar as possible and those of different groups as dissimilar as possible [22]. To ensure that the distributions of the covariates were similar and no covariate had undue influence on the groupings, the covariates were normalised and the SSRs were log transformed prior to normalising. The Calinski-Harabasz index identified the optimal number of groups [23].

## Results

Separations were excluded if the area code was non-spatial, a remote island or had an average annual population of fewer than 5 males aged 40 or more. For radical prostatectomy, separations were excluded for non-spatial residential codes of "offshore, migratory or shipping" (n = 46, 0.06%) or of either "no usual address" or "undefined" (n = 12, 0.01%) and because their residential area did not meet the minimum population threshold (n = 28, 0.03%). For LDR brachytherapy and HDR brachytherapy fewer than 10 separations were excluded for having a non-spatial residential code or the residential code did not meet the population threshold (respectively <0.1% and <0.6%). Fewer than 10 separations for any type of treatment were excluded because the residential location was coded as a remote island. Values less than 10 are considered disclosive and cannot be published.

In total, 94 777 hospital separations for interventional treatments of prostate cancer occurred in 2007–2016 in Australia, of which 83 101 (88%, age-standardised separation rate (95% confidence interval) 15.2 (15.1–15.3) per 10 000 person-years) were radical prostatectomy, 10 029 (11%, 1.87 (1.83–1.91) per 10 000 person-years) were LDR brachytherapy and 1647 (2%, 0.31 (0.30–0.32) per 10 000 person-years) were HDR brachytherapy. These percentages were broadly similar across the States (Table 1), although there was some variation in smaller States and Territories.

**Table 1. Average annual counts and age-standardised separation rates (/10 000 person-years) and 95% confidence intervals (CI) for prostate cancer treatment in Australian males aged 40 years and over between 2007 and 2016.** Percentages of the observed separations for each treatment for prostate cancer are also provided. The Australian standard age distribution 2001 was used for age-standardisation [30].

| | Total | NSW[§] | Victoria | Queensland | WA[¶] | SA[††] | Tasmania | ACT[‡‡] | NT[§§] |
|---|---|---|---|---|---|---|---|---|---|
| **Treatments** | | | | | | | | | |
| Radical prostatectomy | | | | | | | | | |
| Counts (%) | 8310 (88%) | 2918 (89%) | 1933 (87%) | 1718 (89%) | 836 (90%) | 525 (77%) | 186 (83%) | 167 (95%) | 28 (83%) |
| Age standardised rate | 15.2 | 16.3 | 14.5 | 15.7 | 15.1 | 12.2 | 12.8 | 21.7 | 6.3 |
| (95% CI) | (15.1, 15.3) | (16.2, 16.5) | (14.3, 14.7) | (15.5, 16.0) | (14.8, 15.4) | (11.9, 12.6) | (12.2, 13.4) | (20.7, 22.8) | (5.5, 7.1) |
| LDR[†] brachytherapy | | | | | | | | | |
| Counts (%) | 1003 (11%) | 275 (8%) | 269 (12%) | 168 (9%) | 95 (10%) | 152 (22%) | 32 (14%) | 6 (3%) | 5 (16%) |
| Age standardised rate | 1.87 | 1.56 | 2.05 | 1.58 | 1.83 | 3.55 | 2.20 | 0.81 | 1.21 |
| (95% CI) | (1.83, 1.91) | (1.51, 1.63) | (1.97, 2.13) | (1.50, 1.66) | (1.71, 1.95) | (3.38, 3.74) | (1.96, 2.46) | (0.62, 1.05) | (0.89, 1.62) |
| HDR[‡] brachytherapy | | | | | | | | | |
| Counts (%) | 165 (2%) | 86 (3%) | 17 (1%) | 46 (2%) | 2 (<1%) | 4 (1%) | 6 (3%) | 2 (1%) | <1 (1%) |
| Age standardised rate | 0.31 | 0.50 | 0.13 | 0.43 | 0.05 | 0.10 | 0.47 | 0.29 | 0.10 |
| (95% CI) | (0.30, 0.33) | (0.46, 0.53) | (0.11, 0.15) | (0.39, 0.47) | (0.03, 0.07) | (0.07, 0.14) | (0.36, 0.60) | (0.18, 0.45) | (0.03, 27) |
| **Total** | 9478 | 3279 | 2219 | 1933 | 933 | 681 | 224 | 175 | 34 |

[†] LDR: Low dose rate

[‡] HDR: High dose rate

[§] NSW: New South Wales

[¶] WA: Western Australia

[††] SA: South Australia

[‡‡] ACT: Australian Capital Territory

[§§] NT: Northern Territory

## Area-level associations

The modelled separation rates for each treatment varied substantially by State and Territory, area disadvantage and remoteness (p < 0.01 for all treatment types and covariates, Fig 1). Rates for radical prostatectomy were highest in the Australian Capital Territory (18.7/10 000 person-years, 95% CI: 17.7–19.8) and lowest in the Northern Territory (6.8/10 000 person-years, 95% CI: 5.7–7.9). Rates of radical prostatectomy tended to increase with increasing area-level socioeconomic advantage and were lower in remote areas (12.2/10 000 person-years, 95% CI: 10.8–13.5) compared with other remoteness categories.

Separation rates for LDR brachytherapy differed by State and increased with increasing area-level socioeconomic advantage but varied little with remoteness category. Separation rates for HDR brachytherapy varied substantially with highest rates in New South Wales (5.0/100 000 person-years, 95% CI: 4.4–5.6), Queensland (4.9/100 000 person-years, 95% CI: 4.3–5.4), Tasmania (6.4/100 000 person-years, 95% CI: 4.1–8.8) and more socioeconomically advantaged areas (most advantaged: 5.0/100 000 person-years, 95% CI: 4.3–5.7). The wide, overlapping confidence intervals suggest no evidence of a difference by remoteness.

## Spatial modelling

There was strong evidence of spatial variation across Australia for all treatments (p < 0.001). In the maps of SSRs (Fig 2), yellow areas had separation rates similar in magnitude to the national average, blue areas had lower separation rates and red areas had higher separation rates compared to the national average for that treatment.

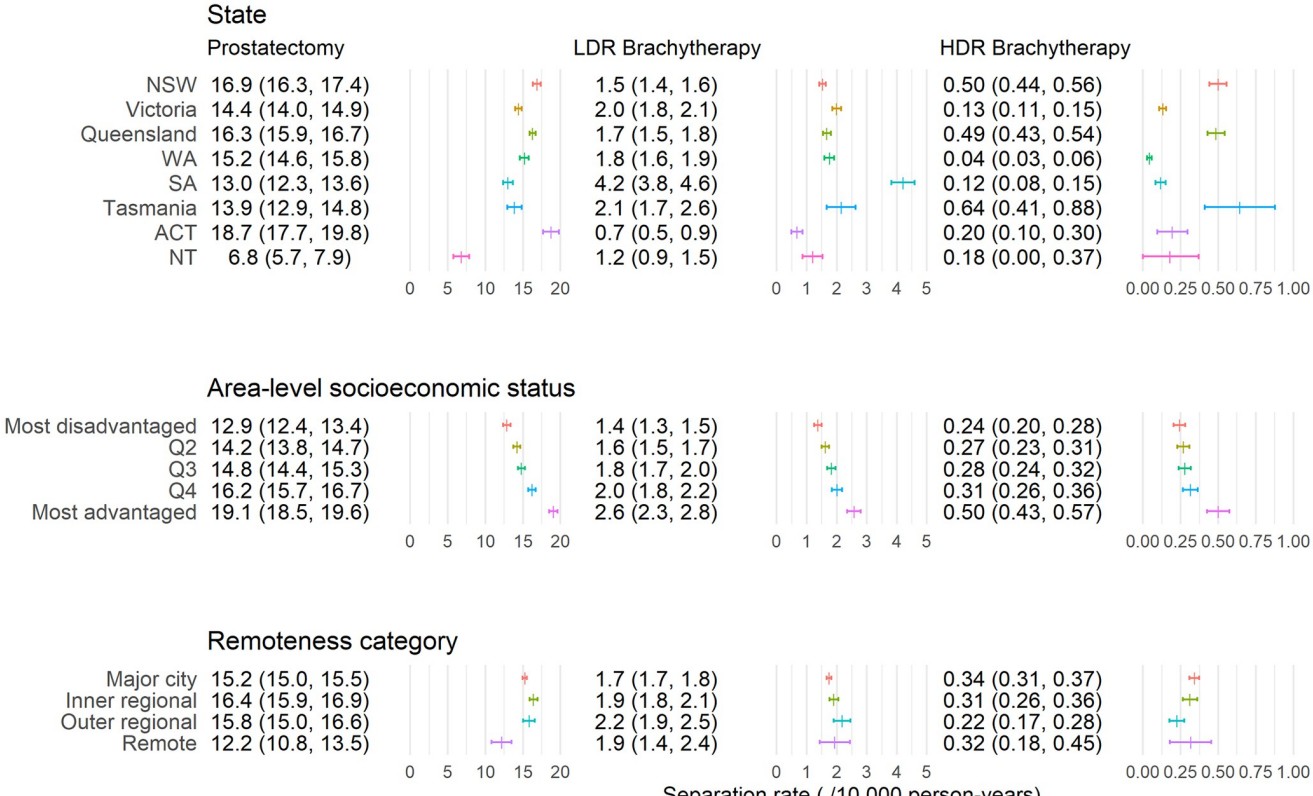

**Fig 1. Age-adjusted marginal separation rates (95% confidence intervals) for men aged 40 and over by State or Territory, area-level socioeconomic quintile and remoteness category for each treatment.** "Prostatectomy" refers to radical prostatectomy. Abbreviations: high dose rate (HDR), low dose rate (LDR), New South Wales (NSW), Western Australia (WA), South Australia (SA), Australian Capital Territory (ACT) and Northern Territory (NT). Note that the horizontal axis scale varies by treatment type.

Remote areas in the north and west of the country had lower than average separation rates for all treatments (Fig 2). For radical prostatectomy, areas with higher SSRs tended to be in eastern areas of mainland Australia. The geographic distribution of SSRs for HDR brachytherapy was similar to that of radical prostatectomy, while that of LDR brachytherapy contrasted markedly. However, this correlation was regionalised, resulting in small correlation coefficients (absolute values less than 0.3) overall. Area-level SIRs for prostate cancer were correlated with SSRs for prostatectomy (0.64).

Three groupings of areas with similar SSRs, SIRs and EHRs were identified through statistical clustering and were labelled the "higher prostatectomy", "high EHR" and "LDR brachytherapy" groups (Fig 3, S1 Fig in S1 Appendix and S4 Table in S1 Appendix), reflecting the prominent characteristic of each group. Areas in the higher prostatectomy group (875 areas, 41%) also tended to have higher prostate cancer incidence and higher separations for HDR brachytherapy. Areas in the LDR brachytherapy group (534 areas, 25%) tended to have low SSRs for HDR brachytherapy. Areas in the high EHR group (739 areas, 34%) had low prostate cancer incidence and low separation rates for each of the three treatments. Remote areas and those with the greatest area disadvantage tended to be in the high EHR group.

As area-level socioeconomic advantage increased, the proportion of areas in the higher prostatectomy group increased and the proportion of areas in the high EHR group generally

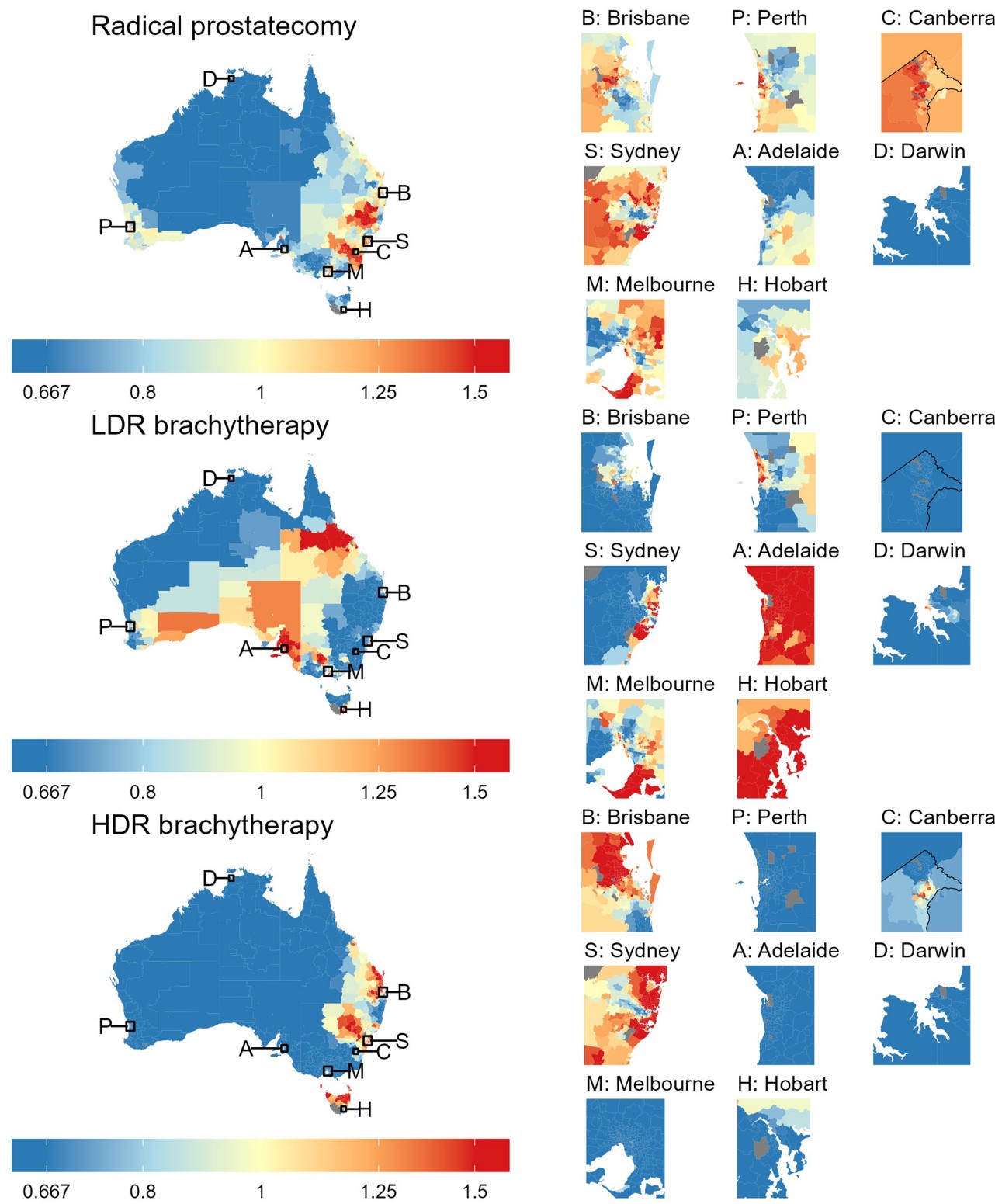

**Fig 2. Maps of the standardised separation ratios (SSRs) across Australia for each treatment.** Treatments were radical prostatectomy (top), low dose rate (LDR) brachytherapy (middle) and high dose rate (HDR) brachytherapy (bottom). Insets show the capital city regions for each State and Territory, which are identified on the main map using their initial. Note that the insets are not of equal dimension.

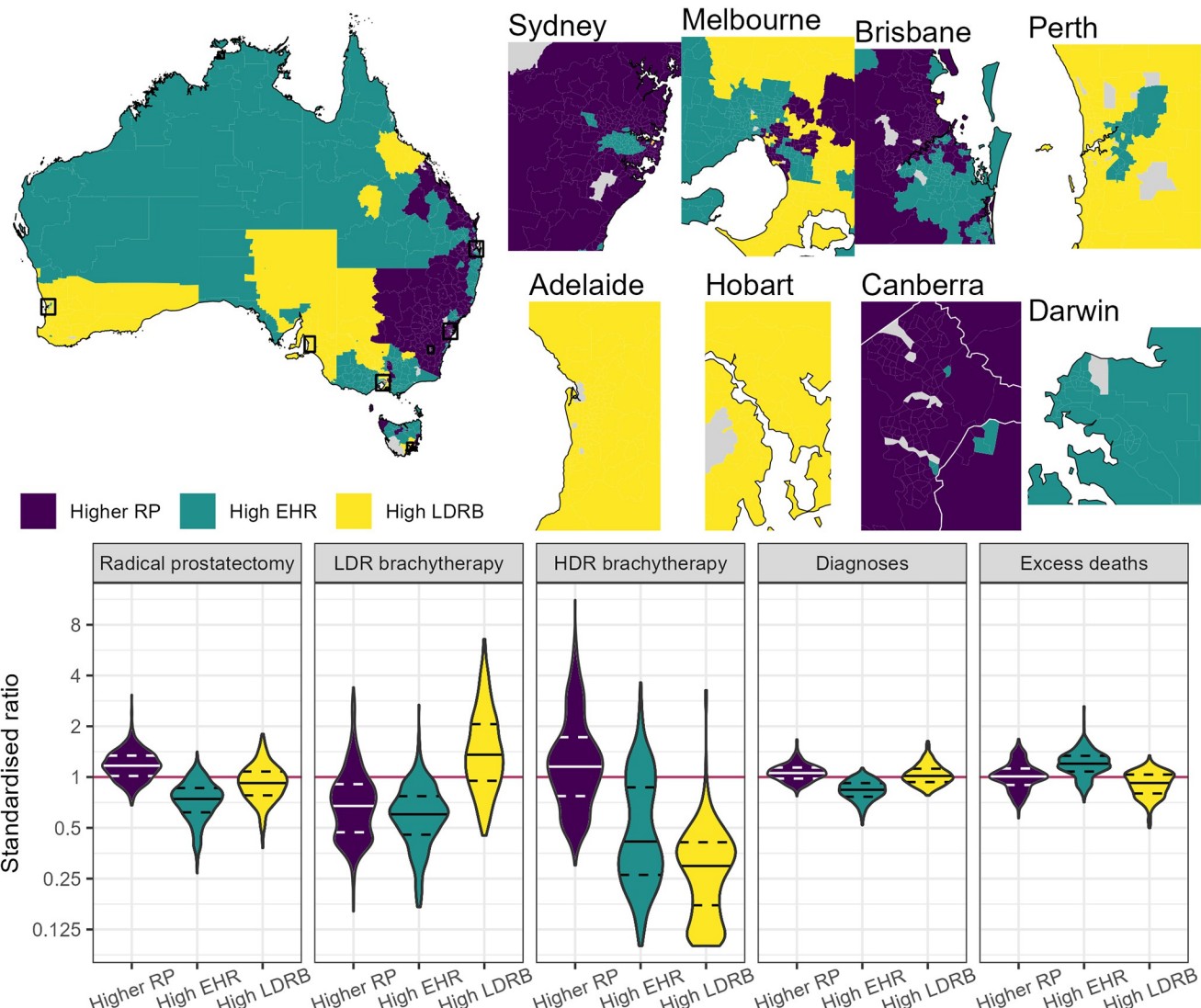

**Fig 3. The geographic distribution and characteristics of the statistical groups.** Maps showing the geographic distribution of the three statistical groups (indicated by colour), insets showing the statistical groups for the State and Territory capital cities and violin plots showing the distribution of standardised rates for each of the treatments, diagnoses and excess deaths by statistical group. The groups have been labelled "higher RP", "high EHR" and "high LDRB" reflecting the key characteristic of each, where RP stands for "radical prostatectomy" and LDRB stands for "low dose rate brachytherapy". The solid black lines in the violins indicate each group's median, the dashed lines indicate the 80th percentiles and the red line at 1 highlights the national average. Note that in the Canberra inset, the border for the Australian Capital Territory is shown in white.

decreased (S4 Table in S1 Appendix). With increasing remoteness, the proportion of areas in the higher prostatectomy group decreased and the proportion of areas in the high EHR group increased. Group allocation was highly correlated with State or Territory.

## Discussion

Our study provided strong evidence of geographic variation in rates of interventional prostate cancer treatments in public and private hospitals across Australia. Correlations between geographical patterns were apparent and areas could be characterised by their rates for particular treatments.

The variation in the separation rates reported in this study reflect variations in both the incidence of prostate cancer in each area and clinical management decisions taken by patients and their health care providers. For men with localised prostate cancer there are often theoretical options available for primary treatment, but distance to treatment centres or financial concerns and may limit these options in practice. Additionally, management decisions are influenced by age, stage/grade of disease, socioeconomic factors including education and financial resources, advice from diagnosing or primary clinicians, potential side effects and comorbidities such as obesity or cardiovascular disease [5]. Finally, a man's decision will be influenced by his life context, beliefs about health, the sociocultural context in which he lives and his personal experiences with cancer, often resulting in a preference for surgery [24].

Regional differences in treatment rates may be confounded by geographic differences in stage and grade. The Australian and New Zealand Prostate Cancer Outcomes Registry has reported small differences in risk groups by State or Territory [11], but these differences are unlikely to explain the larger differences in separation rates observed in this study [11]. Remoteness and area disadvantage have been associated with more advanced stage at diagnosis [4], which may explain the geographical patterns in rates of radical prostatectomy but not in brachytherapy.

The "high EHR" group may comprise areas with low testing rates, resulting in lower incidence of early disease, a higher *proportion* of advanced disease, and thus poorer survival. The hypothesis that areas in this group tend to have higher proportions of advanced disease is also consistent with the low SSRs of areas in this group, since men with regional or metastatic disease are more likely to receive androgen deprivation therapy and/or chemotherapy [11]. The large proportion of remote areas that are in this group may be indicative of poorer access to urological and radiation therapy services. Early diagnosis of prostate cancer is less likely in areas with limited access to general practitioners or urologists, resulting in a lower incidence of early-stage disease, and thus reduced use of interventional procedures such as surgery or radiation therapy.

Geographic differences in treatment rates likely reflect variation in access to and availability of therapies [4]. Differences between States and Territories in treatment rates for brachytherapy may reflect the availability of the technology in certain institutions and the specialist expertise of the clinicians trained in providing this treatment [25]. Services including radiation therapy facilities and specialist care in regional hospitals need to be expanded so that care is more accessible to more men outside major cities [26]. Additionally, the important role of clinicians discussing and advocating for specific treatments with their patients may also impact geographical patterns.

Prostatectomy is performed on men with localised cancer. It is likely that areas with higher rates of PSA testing or prostate biopsies would be associated with higher rates of prostatectomy. While there is some correspondence between areas with high SSRs observed here and areas with previously published high biopsy rates [3], a formal ecological analysis assessing correlations between testing, diagnostic practices and treatment patterns has not been undertaken.

Separation rates decreased with increasing area-level socioeconomic disadvantage. This may reflect the high out-of-pocket costs associated with treatment [27]. Median out-of-pocket costs for treatment of prostate cancer have been found to be $AU 5000, but costs varied depending on type of treatment and State or Territory and were higher for men with private health insurance [27]. The gradient in separation rates by area-level socioeconomic quintile is stronger for radical prostatectomy and there is evidence that this type of surgery is more likely to be conducted in Australian private hospitals than in public hospitals and men with private

health insurance are more likely to have a radical prostatectomy than men without private health insurance [8, 27, 28].

Comorbidities are known to influence treatment decisions; [8, 28] for example, men with very limited physical functioning and obese men are less likely to have a radical prostatectomy, after adjusting for other comorbidity, age and socioeconomic factors [8]. This is consistent with our results, since the prevalence of obesity and other morbidity tends to be greater in regional and remote areas and more disadvantaged areas [29], fewer of which were in the higher prostatectomy group.

## Limitations

As an ecological study, this study cannot definitively identify the reasons for variation in patterns of treatment. Hospital separations data included interventional treatments only and not active surveillance, watchful waiting, external beam radiotherapy, androgen deprivation therapy, chemotherapy, or combination therapy. Critically we have *no* data on external beam radiation therapy, the second most common form of active treatment for prostate cancer. Overall, these gaps in the data may explain why treatment rates are apparently low in some areas, particularly for radical prostatectomy. It was not possible to determine whether men had combinations of treatments, nor whether the treatment was primary treatment or a delayed intervention after a period of active surveillance. When converting between geography standards, in the absence of more specific correspondence matrices, the matrix used was for all persons. The SSRs and EHRs were modelled over 2007–2016 and the modelling may have included men who were diagnosed prior to 2007, while the SIRs were from 2012–2016. Analyses of spatially aggregated data may be sensitive to how the area units are defined. SA2s are designed to aggregate populations who interact socially and economically, and represent the optimal balance between spatial resolution and privacy concerns.

Nationally, the mix of interventional treatments changed during the study period [6] and spatiotemporal modelling of the data should be conducted in the future.

## Conclusions

We found marked geographical variation in population rates for interventional treatments for prostate cancer across Australia. Geographical variation in age-standardised separation rates for these treatment options was consistent with the geographical variation of factors associated with treatment decisions such as stage at diagnosis, comorbidities, financial situation, geographic proximity and socioeconomic disadvantage. Separation rates for the treatments studied decreased with increasing area-level socioeconomic disadvantage and rates of radical prostatectomy were lower in remote areas. Statistical clustering identified a set of areas with low age-adjusted standardised separation rate ratios and low standardised incidence ratios but high excess death ratios. Improving access to services and facilities may improve geographical disparities in cancer outcomes.

## Supporting information

**S1 Appendix. Supplementary methods and results.** This file includes additional results mentioned in the text of the manuscript as well as the complete model outputs for the "Area-level associations" section and a detailed description of the methods and sensitivity analysis. (PDF)

**S2 Appendix. Modelled estimates.** This file includes the standardised separation rate ratios presented in Figs 2 and 3.
(XLSX)

## Acknowledgments

The authors appreciate the work of staff of the Australian Institute of Health and Welfare in collecting and providing the data for these analyses through the National Hospital Morbidity Database.

## Author Contributions

**Conceptualization:** Jessica K. Cameron, Jeff Dunn, Suzanne K. Chambers, Peter D. Baade, David P. Smith.

**Data curation:** Jessica K. Cameron.

**Formal analysis:** Jessica K. Cameron, Upeksha Chandrasiri, Jeremy Millar, Mark Frydenberg, Peter D. Baade, David P. Smith.

**Funding acquisition:** Suzanne K. Chambers, Peter D. Baade, David P. Smith.

**Investigation:** Jessica K. Cameron, Peter D. Baade.

**Methodology:** Jessica K. Cameron, Susanna Cramb, Kerrie Mengersen, Peter D. Baade.

**Project administration:** Peter D. Baade, David P. Smith.

**Software:** Jessica K. Cameron, Susanna Cramb.

**Supervision:** Peter D. Baade, David P. Smith.

**Validation:** Jessica K. Cameron.

**Visualization:** Jessica K. Cameron, Kerrie Mengersen, Peter D. Baade.

**Writing – original draft:** Jessica K. Cameron, Upeksha Chandrasiri.

**Writing – review & editing:** Jessica K. Cameron, Jeremy Millar, Joanne F. Aitken, Susanna Cramb, Jeff Dunn, Mark Frydenberg, Prem Rashid, Kerrie Mengersen, Peter D. Baade, David P. Smith.

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
