## [Decision Letter · Decision Letter 0]

21 Aug 2023

PONE-D-23-18591Disease mapping: geographic differences in population rates of interventional treatment for prostate cancer in AustraliaPLOS ONE

Dear Dr. Cameron,

Thank you for submitting your manuscript to PLOS ONE. After careful consideration, we feel that it has merit but does not fully meet PLOS ONE’s publication criteria as it currently stands. Therefore, we invite you to submit a revised version of the manuscript that addresses the points raised during the review process.

We look forward to receiving your revised manuscript.

Kind regards,

Marianna De Camargo Cancela, DDS, MSc, PhD

Academic Editor

PLOS ONE

2. We noted in your submission details that a portion of your manuscript may have been presented or published elsewhere.

[A preprint of the manuscript is available on medrxiv and this has been uploaded with this submission.

https://www.medrxiv.org/content/10.1101/2023.06.12.23291310v1

The data provided in the Supporting Information will be made available through the Australian Cancer Atlas (https://atlas.cancer.org.au/) later in 2023. If this is a problem for PlosOne, I will remove the Supporting Information file.

Otherwise, the results, data and figures in the manuscript have not been published elsewhere at this time.]

Please clarify whether this publication was peer-reviewed and formally published. If this work was previously peer-reviewed and published, in the cover letter please provide the reason that this work does not constitute dual publication and should be included in the current manuscript.

4. We note that Figures 2 and 3 in your submission contain [map/satellite] images which may be copyrighted. All PLOS content is published under the Creative Commons Attribution License (CC BY 4.0), which means that the manuscript, images, and Supporting Information files will be freely available online, and any third party is permitted to access, download, copy, distribute, and use these materials in any way, even commercially, with proper attribution. For these reasons, we cannot publish previously copyrighted maps or satellite images created using proprietary data, such as Google software (Google Maps, Street View, and Earth). For more information, see our copyright guidelines: http://journals.plos.org/plosone/s/licenses-and-copyright.

a. You may seek permission from the original copyright holder of Figures 2 and 3 to publish the content specifically under the CC BY 4.0 license. 

Reviewers' comments:

Reviewer's Responses to Questions

**Comments to the Author**

1. Is the manuscript technically sound, and do the data support the conclusions?

Reviewer #1: Yes

Reviewer #2: Partly

2. Has the statistical analysis been performed appropriately and rigorously? 

Reviewer #1: Yes

Reviewer #2: No

3. Have the authors made all data underlying the findings in their manuscript fully available?

Reviewer #1: Yes

Reviewer #2: No

4. Is the manuscript presented in an intelligible fashion and written in standard English?

Reviewer #1: Yes

Reviewer #2: Yes

5. Review Comments to the Author

Reviewer #1: The authors present a paper about "sease mapping: geographic differences in population rates of interventional treatment for prostate cancer in Australia".

The topic itself has a major interest rather than just for Australia because it looks for connections among clinical outcomes and social/geographic factors.

The process of data search and analisys has been clearly presented by the researchers and the limitations of this study have also been adequately addressed.

Reviewer #2: Review of: Disease mapping: geographic differences in population rates of interventional treatment for prostate cancer in Australia.

This is a well written paper on a themed topic. This ecological study used various statistical methods to compare procedures done following the diagnosis of prostate cancer by regions.

Major concerns:

Backward elimination method: The authors used elimination method but did not report the final covariates that were included in the models. Elimination methods (although still popular and wrongly so) may introduce wrong results, as these often tend to underestimate the standard errors of the coefficient estimates, making the confidence intervals too narrow and the p values too low, which may lead to overfitting, and this creates a false confidence in the final model.

Another clear disadvantage of such methods is that once a variable is dropped, it is not re-entered. However, a dropped variable may sometimes become significant in a full model that did not follow the elimination method.

I advise the authors to show the results of the full model that included all covariates and the one that followed the elimination method. The covariates remaining after the elimination must be reported in this paper.

Although the analysis used age-standardised counts, these were based on aggregated data. Such ecological studies fail to properly adjust for important confounders such as age. Confounding by age is very likely in this study because often men residing in remote and rural locations may be diagnosed at an older age and often at a later stage than those residing in metropolitan regions. Older age and more advanced disease are correlated with less aggressive management. As the authors have correctly stated in their introduction that age at the diagnosis and stage of the disease are critical factors that influence the management. Age and stage of the disease are important confounders that are not accounted for in this ecological study.

Competing risk: In such cases where the disease is more advanced and the patient is older, death is a competing risk when considering management. If mortality data (even aggregated) are available, the possible competing risk of mortality can be assessed using Bayesian models with spatial random effects for the clustered data.

Although spatial analysis is informative and is appropriate in ecological studies, such analyses have some limitations often due to the set boundaries by regions. Possible problems that could arise from this relate to the shape effect and/or the edge effect. The authors should relate to these and to further discuss in limitations.

Minor concerns:

In the introduction, instead of providing crude number of cases, please report the age-standardised incidence rate per population. Please also report if the incidence has changed over time as you do say that mortality has decreased over time.

The lower incidence of cancer in remote areas may have been confounded by the competing risk of death. Also the stage of cancer at diagnosis could have differed by region.

6. PLOS authors have the option to publish the peer review history of their article (what does this mean?). If published, this will include your full peer review and any attached files.

Reviewer #1: No

Reviewer #2: **Yes: **George Mnatzaganian

---

## [Author Response · Author response to Decision Letter 0]

15 Sep 2023

We have ensured that PLOS ONE’s style requirements are met.

2. We noted in your submission details that a portion of your manuscript may have been presented or published elsewhere.

Please clarify whether this publication was peer-reviewed and formally published. If this work was previously peer-reviewed and published, in the cover letter please provide the reason that this work does not constitute dual publication and should be included in the current manuscript.

The manuscript was uploaded to medrxiv, but has not been peer reviewed or formally published elsewhere.

The ethics statement has been removed from the Disclosure section. The ethics statement remains in the Methods section.

4. We note that Figures 2 and 3 in your submission contain [map/satellite] images which may be copyrighted. All PLOS content is published under the Creative Commons Attribution License (CC BY 4.0), which means that the manuscript, images, and Supporting Information files will be freely available online, and any third party is permitted to access, download, copy, distribute, and use these materials in any way, even commercially, with proper attribution. For these reasons, we cannot publish previously copyrighted maps or satellite images created using proprietary data, such as Google software (Google Maps, Street View, and Earth). For more information, see our copyright guidelines: http://journals.plos.org/plosone/s/licenses-and-copyright.

The maps in this manuscript were constructed by the corresponding author (JC) using public domain mapping data and plotting software. Satellite images were not used. The modelled estimates plotted in the maps were generated by the analyses carried out for this study. The data describing the shapes of the areas (the “shapefiles”) are publicly available from the Australian Bureau of Statistics for use without license. The maps were constructed using freeware: the ggplot2 package and R.

Reviewers' comments:

Reviewer's Responses to Questions

Comments to the Author

1. Is the manuscript technically sound, and do the data support the conclusions?

Reviewer #1: Yes

Reviewer #2: Partly

This study analysed population-level administrative data, and so controls, replication and sample size are not applicable. We have addressed Reviewer 2’s comments in detail in the “Review Comments to the Author” section below.

2. Has the statistical analysis been performed appropriately and rigorously?

Reviewer #1: Yes

Reviewer #2: No

We have addressed Reviewer 2’s comments in detail in the “Review Comments to the Author” section below.

3. Have the authors made all data underlying the findings in their manuscript fully available?

Reviewer #1: Yes

Reviewer #2: No

The standardised separation rate ratios used to construct the figures and maps have been supplied as supplementary material. The standardised incidence ratios and excess hazard ratios used in the clustering are freely available to download from the Australian Cancer Atlas, as described in the Methods (line 137). We have added the model coefficients, standard errors and 95% confidence intervals for the “Area-level associations” section to the Supplementary material.

4. Is the manuscript presented in an intelligible fashion and written in standard English?

Reviewer #1: Yes

Reviewer #2: Yes

Thank you.

5. Review Comments to the Author

Reviewer #1: The authors present a paper about "sease mapping: geographic differences in population rates of interventional treatment for prostate cancer in Australia".

The topic itself has a major interest rather than just for Australia because it looks for connections among clinical outcomes and social/geographic factors.

The process of data search and analisys has been clearly presented by the researchers and the limitations of this study have also been adequately addressed.

Thank you for your encouraging comments and taking the time to review the manuscript.

Reviewer #2: Review of: Disease mapping: geographic differences in population rates of interventional treatment for prostate cancer in Australia.

This is a well written paper on a themed topic. This ecological study used various statistical methods to compare procedures done following the diagnosis of prostate cancer by regions.

Major concerns:

Backward elimination method: The authors used elimination method but did not report the final covariates that were included in the models. Elimination methods (although still popular and wrongly so) may introduce wrong results, as these often tend to underestimate the standard errors of the coefficient estimates, making the confidence intervals too narrow and the p values too low, which may lead to overfitting, and this creates a false confidence in the final model.

Another clear disadvantage of such methods is that once a variable is dropped, it is not re-entered. However, a dropped variable may sometimes become significant in a full model that did not follow the elimination method.

I advise the authors to show the results of the full model that included all covariates and the one that followed the elimination method. The covariates remaining after the elimination must be reported in this paper.

None of the covariates were dropped from the final model and the results shown in Figure 1 are the results for the full model. The Methods section (line 151-154) has been edited to ensure clarity and now reads:

“All covariates were retained in the multivariable model. Likelihood ratio tests were used initially to test for univariable associations and to determine the strength of evidence for the association in the final, fully-adjusted model.”

We have also included the results of the likelihood ratio tests for the fully-adjusted model in the first sentence of the “Area-level associations” section of the Results: “p < 0.01 for all treatment types and covariates” (line 199).

Although the analysis used age-standardised counts, these were based on aggregated data. Such ecological studies fail to properly adjust for important confounders such as age. Confounding by age is very likely in this study because often men residing in remote and rural locations may be diagnosed at an older age and often at a later stage than those residing in metropolitan regions. Older age and more advanced disease are correlated with less aggressive management. As the authors have correctly stated in their introduction that age at the diagnosis and stage of the disease are critical factors that influence the management. Age and stage of the disease are important confounders that are not accounted for in this ecological study.

The data custodians did not supply small area data by single-year age to conform with privacy and data confidentiality requirements, hence it is not possible to carry out this study with single-year age data. However, recognising the importance of age, we obtained data aggregated into ten-year age groups. All models were adjusted for these age groups. The GLMMs described in the “Area-level associations” section were directly adjusted by including age as a covariate, as described in the Methods section (line 149-151) as follows:

“Generalised linear models were built to model area-level count data between 2007 and 2016 for each procedure with the following covariates: age group,…”

The spatial estimates were indirectly adjusted for age, as described in the “Spatial modelling” section of the Methods (line 161-163):

“Briefly, counts of separations in the period 2007 to 2016 for each treatment were modelled separately, offset by the age-adjusted expected counts.”

The indirect adjustment provides standardised separation rate ratios which facilitate meaningful comparisons of the outcomes for large numbers of subpopulations.

Stage for prostate cancer is not consistently collected or reported across Australia, hence, it is not possible to adjust for stage in these models. Moreover, the goal of this study was to report geographic variation in treatment rates (see the goals stated in the last paragraph of the Introduction, page 5). Given the geographical differences in the reporting of stage for prostate cancer, including stage in the model will complicate the interpretation of the results.

However, the possible impact of disease stage on the spatial variation in treatment rates is discussed at length in the Discussion, particularly the third paragraph of the Discussion (page 14, line 277), which begins “Regional differences in treatment rates may be confounded by geographic differences in stage and grade.”

Competing risk: In such cases where the disease is more advanced and the patient is older, death is a competing risk when considering management. If mortality data (even aggregated) are available, the possible competing risk of mortality can be assessed using Bayesian models with spatial random effects for the clustered data.

Thank you for this interesting suggestion. Because men may be treated for prostate cancer months or years after their diagnosis, direct comparisons cannot be made with cancer survival data. Attempting to adjust the models using the excess hazard ratios would therefore be problematic. Moreover, the goal of the study was to describe the spatial patterns and adjusting for stage would complicate the interpretation. The modelling was adjusted for age and therefore age-related differences in management and mortality.

K-means clustering was carried out to understand ecological correlations between treatment rates, incidence and excess mortality. The clustering identified a group of areas with higher rates of excess mortality and lower rates of both radical prostatectomy and low dose rate brachytherapy, compared with the Australian average. This result was commented on in terms of disease stage in the Discussion (line 283-291) as follows:

‘The “high EHR” group may comprise areas with low testing rates, resulting in lower incidence of early disease, a higher proportion of advanced disease, and thus poorer survival. The hypothesis that areas in this group tend to have higher proportions of advanced disease is also consistent with the low SSRs of areas in this group, since men with regional or metastatic disease are more likely to receive androgen deprivation therapy and/or chemotherapy.[11] The large proportion of remote areas that are in this group may be indicative of poorer access to urological and radiation therapy services. Early diagnosis of prostate cancer is less likely in areas with limited access to general practitioners or urologists, resulting in a lower incidence of early-stage disease, and thus reduced use of interventional procedures such as surgery or radiation therapy.’

Although spatial analysis is informative and is appropriate in ecological studies, such analyses have some limitations often due to the set boundaries by regions. Possible problems that could arise from this relate to the shape effect and/or the edge effect. The authors should relate to these and to further discuss in limitations.

We have added the following to the Limitations (line 329-332):

“Analyses of spatially aggregated data may be sensitive to how the area units are defined. SA2s are designed to aggregate populations who interact socially and economically, and represent the optimal balance between spatial resolution and privacy concerns.”

Minor concerns:

In the introduction, instead of providing crude number of cases, please report the age-standardised incidence rate per population. Please also report if the incidence has changed over time as you do say that mortality has decreased over time.

The age-standardised incidence rates have been added to the first paragraph of the Introduction (line 70-72), as follows:

“Age-standardised incidence rates have increased to 151 cases per 100,000 persons-years in 2022 from 80 cases per 100,000 person-years in 1982.[1]”

The lower incidence of cancer in remote areas may have been confounded by the competing risk of death. Also the stage of cancer at diagnosis could have differed by region.

This study aimed to describe treatment rates for prostate cancer, so incidence rates are outside the scope of this paper. Stage at diagnosis does differ geographically and was discussed in the third paragraph of the Discussion (line 277-282), as follows:

“Regional differences in treatment rates may be confounded by geographic differences in stage and grade. The Australian and New Zealand Prostate Cancer Outcomes Registry has reported small differences in risk groups by State or Territory,[11] but these differences are unlikely to explain the larger differences in separation rates observed in this study.[11] Remoteness and area disadvantage have been associated with more advanced stage at diagnosis,[4] which may explain the geographical patterns in rates of radical prostatectomy but not in brachytherapy.”

---

## [Editor Report · Decision Letter 1]

24 Oct 2023

Disease mapping: geographic differences in population rates of interventional treatment for prostate cancer in Australia

PONE-D-23-18591R1

Dear Dr. Cameron,

We’re pleased to inform you that your manuscript has been judged scientifically suitable for publication and will be formally accepted for publication once it meets all outstanding technical requirements.

Kind regards,

Marianna De Camargo Cancela, DDS, MSc, PhD

Academic Editor

PLOS ONE

---

## [Editor Report · Acceptance letter]

3 Nov 2023

PONE-D-23-18591R1 

Disease mapping: geographic differences in population rates of interventional treatment for prostate cancer in Australia 

Dear Dr. Cameron:

I'm pleased to inform you that your manuscript has been deemed suitable for publication in PLOS ONE. Congratulations! Your manuscript is now with our production department. 

Kind regards, 

on behalf of

Dr Marianna De Camargo Cancela 

Academic Editor

PLOS ONE